# Long-Term Environmental Hypoxia Exposure and Haematopoietic Prolyl Hydroxylase-1 Deletion Do Not Impact Experimental Crohn’s Like Ileitis

**DOI:** 10.3390/biology10090887

**Published:** 2021-09-08

**Authors:** Cara De Galan, Martine De Vos, Pieter Hindryckx, Debby Laukens, Sophie Van Welden

**Affiliations:** 1Department of Internal Medicine and Paediatrics, Ghent University, 9000 Ghent, Belgium; cara.degalan@ugent.be (C.D.G.); martine.devos@ugent.be (M.D.V.); pieter.hindryckx@uzgent.be (P.H.); debby.laukens@ugent.be (D.L.); 2Ghent Gut Inflammation Group (GGIG), Ghent University, 9000 Ghent, Belgium; 3VIB Centre for Inflammation Research, 9000 Ghent, Belgium; 4Department of Gastroenterology, Ghent University Hospital, 9000 Ghent, Belgium

**Keywords:** Ileal hypoxia, TNF^∆ARE/+^ mice, prolyl hydroxylase 1, immune cell-specific, hypoxia-induced signalling pathways

## Abstract

**Simple Summary:**

Hypoxia-induced signalling represents an important contributor to inflammatory bowel disease (IBD) pathophysiology. However, available data solely focus on colonic inflammation while the primary disease location in Crohn’s disease patients is the terminal ileum. Therefore, we explored the effects of environmental hypoxia and immune cell-specific deletion of oxygen sensor prolyl hydroxylase (PHD) 1 in a Crohn’s like ileitis mouse model. Five-week-old TNF^∆ARE/+^ mice and wildtype (WT) littermates were housed in normoxia (21% O_2_) or hypoxia (8% O_2_) for 10 weeks. Although environmental hypoxia increased both systemic as ileal markers of hypoxia, the body weight evolution in both WT and TNF^∆ARE/+^ mice was not affected. Interestingly, hypoxia did increase circulatory monocytes, ileal mononuclear phagocytes and proinflammatory cytokine expression in WT mice. However, no histological or inflammatory gene expression differences in the ileum could be identified between TNF^∆ARE/+^ mice housed in hypoxia versus normoxia nor between TNF^∆ARE/+^ and WT mice with additional loss of immune cell-specific Phd1 expression. This is the first study showing that long-term environmental hypoxia or haematopoietic Phd1-deletion does not impact experimental ileitis. Therefore, it strongly questions whether targeting hypoxia-induced signalling via currently available PHD inhibitors would exert an immune suppressive effect in IBD patients with ileal inflammation.

**Abstract:**

Environmental hypoxia and hypoxia-induced signalling in the gut influence inflammatory bowel disease pathogenesis, however data is limited to colitis. Hence, we investigated the effect of environmental hypoxia and immune cell-specific deletion of oxygen sensor prolyl hydroxylase (PHD) 1 in a Crohn’s like ileitis mouse model. Therefore, 5-week-old C57/BL6 TNF^∆ARE/+^ mice and wildtype (WT) littermates were housed in normoxia (21% O_2_) or hypoxia (8% O_2_) for 10 weeks. Systemic inflammation was assessed by haematology. Distal ileal hypoxia was evaluated by pimonidazole staining. The ileitis degree was scored on histology, characterized via qPCR and validated in haematopoietic Phd1-deficient TNF^∆ARE/+^ mice. Our results demonstrated that hypoxia did not impact body weight evolution in WT and TNF^∆ARE/+^ mice. Hypoxia increased red blood cell count, haemoglobin, haematocrit and increased pimonidazole intensity in the ileum. Interestingly, hypoxia evoked an increase in circulatory monocytes, ileal mononuclear phagocytes and proinflammatory cytokine expression in WT mice. Despite these alterations, no histological or ileal gene expression differences could be identified between TNF^∆ARE/+^ mice housed in hypoxia versus normoxia nor between haematopoietic Phd1-deficient TNF^∆ARE/+^ and their WT counterparts. Therefore, we demonstrated for the first time that long-term environmental hypoxia or haematopoietic Phd1-deletion does not impact experimental ileitis development.

## 1. Introduction

Inflammatory bowel diseases (IBD) are chronic, relapsing inflammatory disorders of the gastrointestinal tract, comprising Crohn’s disease (CD) and Ulcerative Colitis (UC) [1,2]. The disease is characterized by a complex interplay of genetic, microbial and environmental factors that influence mucosal homeostasis and induce an inadequate immune response [2].

Environmental hypoxia has been increasingly recognized as an important influence factor of IBD development. In particular, high altitude flights and travelling to high altitude regions were associated with an increased risk for the occurrence of IBD flares within 4 weeks of travel [3]. In contrast, a more recent paper demonstrated that hypoxic exposure counteracts colonic inflammation in both CD patients in clinical remission and dextran sodium sulphate (DSS)-treated mice via the activation of autophagy as a result of decreased nucleotide-binding oligomerization domain receptor, pyrin domain containing 3 (NLRP3)/mammalian target of rapamycin (mTOR) binding [2]. Although very interesting, these studies focused solely on colonic inflammation while in approximately two-thirds of CD cases, the primary location of inflammatory lesions is the small intestine, more specifically the terminal ileum [4]. The impact of environmental hypoxia on chronic ileitis onset and course remains elusive.

In addition to hypoxia as an environmental trigger, mucosal hypoxia is an integral component of gut homeostasis and IBD pathology. In normal physiological conditions, the gastrointestinal tract is characterized by a steep oxygen gradient from the anaerobic lumen toward the highly vascularized and oxygen-rich submucosa while tissue hypoxia extends into the mucosa during intestinal inflammation in mouse models of colitis [5,6]. Yet, if and how hypoxia affects ileal homeostasis under physiological conditions has not been studied in detail. To ensure cell survival and to counteract increased tissue hypoxia, oxygen-sensitive prolyl hydroxylases (PHDs: PHD1/PHD2/PHD3) are inhibited, leading to the activation of hypoxia-inducible factors (HIFs) and the modulation of the nuclear factor-κB (NF-κB) pathway [2,7,8]. In turn, this induces the expression of genes involved in cell proliferation, survival, metabolism, barrier protection and angiogenesis. Therefore, boosting the hypoxia-induced signalling pathways holds therapeutic potential [7].

Out of the three PHD isoforms, targeting PHD1 is most likely the best approach. More specifically, our group demonstrated that PHD1 expression is elevated in inflamed colonic biopsies from both UC and CD patients compared with healthy controls [9]. Furthermore, it was shown that germline loss of only the Phd1 isoform protects against colitis in DSS-treated mice, associated with decreased epithelial cell apoptosis and consequent enhancement of intestinal epithelial barrier function [10]. Our group further demonstrated that haematopoietic Phd1-deletion is necessary and sufficient to suppress colonic inflammation, which is partly due to the promotion of macrophage polarization towards the anti-inflammatory M2 phenotype [11]. Since M2 macrophages antagonize intestinal inflammation and are underrepresented in the colon, but also in the ileal lamina propria of CD patients [12,13], immune cell-specific Phd1-deletion might therefore also beneficially impact inflammation in the small intestine. In this study, we aimed to evaluate the effect of environmental hypoxia and haematopoietic Phd1-deficiency during experimental Crohn’s like ileitis.

## 2. Materials and Methods

### 2.1. Animals

The TNF^∆ARE/+^ mice were kindly provided by Dr. George Kollias (Alexander Fleming Biomedical Sciences Research Centre, Vari, Greece). TNF^∆ARE/+^ mice have a targeted disruption of the TNF AU-rich elements (ARE), which are important for TNF mRNA destabilization and translational repression. Therefore, these mice chronically overproduce TNF-α, which results in the spontaneous development of Crohn’s like intestinal inflammation exclusively located at the terminal ileum around 5 weeks of age. The clinical characterization of these mice is based on a severe reduction in weight gain in comparison to that of the healthy littermate controls. Histologically, terminal ileal lesions consist of mucosal abnormalities with intestinal villous blunting, broadening and shortening, as well as mucosal and submucosal infiltration of acute and chronic inflammatory cells [14,15,16].

The heterozygous Phd1^f/+^Vav:cre^+/−^TNF^ΔARE/+^ (*n* = 15), Phd1^f/+^Vav:cre^−/−^TNF^ΔARE/+^ (*n* = 10), Phd1^f/+^Vav:cre^+/−^ (*n* = 12), Phd1^f/+^Vav:cre^−/−^ (*n* = 22) and homozygous Phd1^f/f^Vav:cre^+/−^TNF^ΔARE/+^ (*n* = 7), Phd1^f/f^Vav:cre^−/−^ TNF^ΔARE/+^ (*n* = 7), Phd1^f/f^Vav:cre^+/−^ (*n* = 7) and Phd1^f/f^Vav:cre^−/−^ (*n* = 7) mice were obtained by consecutively crossing TNF^ΔARE/+^ mice with Phd1^f/f^Vav:cre mice. The latter has cre recombinase under the control of the vav promotor and therefore drives Phd1-deletion specifically in all haematopoietic cells [11]. The generation of the floxed Phd1^f/f^ mice has been previously described [17]. The mice used in this Phd1 deletion experiment were male and female on a C57BL/6 background.

All mice were bred and randomized in open cages in a temperature-controlled room at 22 °C with a 12 h/12 h light/dark cycle. The mice had free access to water and commercial chow (mice maintenance chow; Carfill Labofood, Pavan Service, Oud-Turnhout, Belgium). All mice were treated in accordance with the institutional animal health care guidelines, following study approval (ECD2018/66 and ECD2016/30) by the Institutional Review Board at the Faculty of Medicine and Health Sciences of Ghent University.

### 2.2. Environmental Hypoxia Exposure to TNF^Δare/+^ Mice and Wildtype (WT) Littermates

Five-week-old male TNF^∆ARE/+^ mice and their corresponding WT littermate controls were randomized by weight and PCR confirmed genotype. The mean weight differences between all experimental groups (*n* = 7–8) were not more than 0.5 g. Cages were separated into hypoxia and normoxia groups. The hypoxia groups were exposed to environmental hypoxia (8% O_2_) [2] by placing the mice in a hypoxic chamber (Proox Model 110, BioSpheric, Parish, NY, USA), whereas the normoxia groups (21% O_2_) were used as controls. To ensure the same stress level in all mice caused by the sound of nitrogen replacement in the hypoxic chamber, all groups were housed in the same room. Body weight and general well-being was monitored three times a week. One hour before sacrification, the hypoxic and normoxic mice were intraperitoneally injected with pimonidazole (60 mg/kg) (Hypoxyprobe, Burlington, NJ, USA) in the hypoxic chamber or in the normoxic environment, respectively. Pimonidazole binds to thiol containing proteins in the cells with a low oxygen level (or oxygen pressure (pO_2_)) that equals 10 mm Hg at 37 °C, due to a reduction of the nitro group of the imidazole ring during hypoxia. Pimonidazole binding in hypoxic cells was visualized using immunohistochemical staining (see 2.6). Anaesthesia was obtained by intraperitoneally injecting a 100 µL ketamin (100 mg/kg) (Nimatek, Amsterdam, The Netherlands)/xylazin (10 mg/kg) (Kela, Sint-Niklaas, Belgium) solution in the hypoxic chamber for the hypoxic group or in a normoxic environment, followed by retro-orbital blood collection and subsequent euthanasia via cervical dislocation. During dissection, the distal ileum was collected, cut open longitudinally and divided in three equal parts, starting nearest to the caecum to further proceed towards mid-ileum, and were respectively collected for histology, RNA and DNA analyses. The ileal samples for RNA and DNA analyses were snap frozen and stored at −80 °C. Histological samples from the distal ileum were spread out on the histology patch, followed by fixation in 4% formaldehyde for 24 h and embedding in paraffin with the longitudinal side upwards.

### 2.3. Genotyping of the Included Mice

Genotyping of all mice was performed on an ear, tail or toe piece before inclusion in the experiments and validated at sacrification. To confirm the Phd1-deletion in the distal ileum samples, the DNA was isolated using the TRIzol Reagent (Thermofisher, Paisley, UK) according to the manufacturer’s instructions. Briefly, the distal ileum was homogenized in TRIzol and 200 µL chloroform was added, followed by vigorous manual shaking. After a 3 min incubation step, the samples were centrifuged at 12,000 g for 15 min at 4 °C. The DNA-containing interphase was precipitated with 300 µL 100% ethanol and centrifuged for 5 min at 2000 g at 4 °C to pellet the DNA. Afterwards, the DNA pellet was washed twice with 1 mL of 0.1M sodium citrate in 10% ethanol (pH 8.5) and incubated for 30 min with occasional inversion. Next, an additional wash step in 1 mL of 75% ethanol was performed, and after centrifugation at 2000 g for 5 min at 4 °C, the DNA pellet was air-dried and resuspended in 300 µL 8mM NaOH. Resuspended pellets were left overnight at room temperature to improve solubility. Following the DNA extraction, PCR reactions were performed to assess the presence of the cre recombinase and floxed Phd1 alleles using the AllTaq Master mix (Qiagen, Hilden, Germany) together with the respective primers (Appendix A), water and 1 µL DNA of each sample. A seven-step program was run on a PCR machine (Biorad Laboratories, Temse, Belgium). The cycling conditions comprised of 95 °C for 2 min and 15 s, 30 cycles of 58 °C for 15 s, 72 °C for 15 s, 39 cycles of 95 °C for 15 s and finalized with a cycle of 72 °C for 5 min. Consequently, gel electrophoresis was performed on a 2% agarose gel.

### 2.4. Blood Composition Analysis

Three droplets of blood were collected in heparin coated 2 mL tubes (Sarstedt, Numbrecht, Germany) to analyse blood and immune cell composition using a haematology analyzer (Hemavet 950, Erba Diagnostics Mannheim, Germany).

### 2.5. Histological Assessment of Intestinal Inflammation and Tissue Hypoxia

Paraffin-embedded distal ileum sections of 4 µm were stained with haematoxylin and eosin and scored in a blinded fashion by two independent investigators. The histological ileum sections were evaluated for villous destruction and bowel wall influx of inflammatory cells. Villous destruction was scored on a scale of 0–3: 0, normal; 1, thickened villi; 2, blunted villi; 3, destructed villi. Bowel wall infiltration was scored using the following scoring system: 0, normal; 1, infiltrate into muscular layer of mucosa; 2, infiltrate through submucosa with sporadic granulomas; 3, infiltrate through submucosa into muscularis propria (and/or confluent granulomas); 4, regional transmural infiltration; 5, diffuse transmural infiltration and/or crypt abcedation. The sum of the individual components was expressed as the total inflammation score [18].

### 2.6. Pimonidazole Staining

Ileal hypoxia was quantified using the HypoxyProbe Plus 1 kit (HPI, Burlington, MA, USA) targeting the pimonidazole binding in hypoxic cells. Four µm paraffin sections were deparaffinized and rehydrated by serial immersion in xylene and ethanol. Permeabilization of the tissue was performed in Tris-buffered saline (TBS) with 0.05% Tween 20 (Sigma-Aldrich, Saint-Louis, MI, USA). Antigen retrieval was performed in 10 mM citrate (pH = 6.0) (Sigma-Aldrich, Saint-Louis, MI, USA) at 95 °C for 20 min. Tissues were quenched for endogenous peroxidase activity with 3% H_2_O_2_ (Sigma-Aldrich, Saint-Louis, MI, USA) and blocked with phosphate buffered saline (PBS) (Thermofisher, Paisley, UK) with 1% bovine serum albumin (BSA) and 0.09% sodium azide (Sigma-Aldrich, Saint-Louis, MI, USA). The FITC conjugated hypoxyprobe-1 Mab1 antibody (1 µg/mL) (HPI, Burlington, NJ, USA) in blocking buffer was incubated for 60 min at room temperature. The secondary antibody anti-FITC Mab conjugated with horseradish peroxidase (HPI, Burlington, NJ, USA) was incubated for 30 min at room temperature. Peroxidase reaction was induced by the addition of 3,3′-diaminobenzidine (DAB) (Dako, Glostrup, Denmark) for 10 min. Counterstaining was performed with haematoxylin (Thermofisher, Paisley, UK). Pimonidazole intensity was measured from the epithelial layer to the muscularis mucosa using the imaging software Cell (Olympus, Antwerp, Belgium).

### 2.7. RNA Extraction and Quantitative Real-Time Polymerase Chain Reaction

Total RNA was extracted from the distal ileum of the mice (previously snap frozen and stored at -80 °C in RNA later) using the Aurum Total RNA kit (Bio-Rad, Temse, Belgium) and converted to cDNA (SensiFast cDNA synthesis kit, Bioline (GC Biotech BV, Waddinxveen, The Netherlands). The cDNA was diluted to a concentration of 5 ng/µL. Real-time quantification was performed using a SensiMix SYBR kit (Bioline) and 250 nM of forward and reverse primers (BioLegio, Nijmegen, The Netherlands). A two-step program was run on a LightCycler 480 II (Roche, Vilvoorde, Belgium). The cycling conditions comprised 95 °C for 10 min, 45 cycles of 95 °C for 10 s and 60 °C for 1 min. A melting curve analysis confirmed the primer specificities. All reactions were conducted in duplicate, and the data were normalized to the expression of the reference genes succinate dehydrogenase complex subunit (Sdha) and hypoxanthine-guanine phosphoribosyltransferase (Hprt). The efficiency of each primer pair was calculated using a standard curve from reference cDNA. The amplification efficiency was determined using the formula 10^−1/slope^. Final gene expression levels were calculated with the ∆∆Ct method. Primer sequences are given in Appendix A.

### 2.8. Assessment of Circulatory Cytokine Expression

Levels of TNF, CCL2 and IL1β were determined in serum samples from all mice using Luminex bead-based technology according to manufacturer’s guidelines (Bio-rad Laboratories, Temse, Belgium).

### 2.9. Statistics

The data was analysed using IBM SPSS statistics version 27, for Windows (SPSS Inc., Chicago, IL, USA) or the GraphPad Prism software (Graphpad Software Inc., San Diego, CA, USA). In case of normally distributed data, the mean differences between housing environments and genotypes were analysed using an unpaired Student’s *t*-test for independent samples. For non-normal or unknown data distribution, the groups were compared using the non-parametric Mann–Whitney U-test. The differences between all groups independent of genotype or housing environment were analysed using one-way ANOVA. False discovery correction was performed using Sidak correction. Continuous data (body weight and food uptake) were analysed using linear mixed models. Two tailed probabilities were calculated and probability value of *p* < 0.05 was considered statistically significant.

## 3. Results

### 3.1. Environmental Hypoxia Does Not Influence the General Well-Being of WT Mice, nor the Clinical Course of Inflammation in a Model of Crohn’s Like Ileitis

To study the effects of environmental hypoxia on ileal homeostasis under physiological conditions and during the progression of chronic ileitis, WT and TNF^∆ARE/+^ littermates were housed in a normoxic or hypoxic environment (8% O_2_) from the age of 5 up to 15 weeks. During this housing period, no differences in body weight evolution could be observed between housing environment in either genotype. However, a significant drop in body weight was observed in the first 5 days of the experiment in mice housed in hypoxia (Figure 1a). This drop in weight could be attributed to reduced food intake (Appendix A). Next to body weight, no differences in spleen or liver weight could be identified at week 15 between TNF^∆ARE/+^ mice and WT littermates housed in hypoxia versus normoxia (Figure 1b,c).

### 3.2. Environmental Hypoxia Increases Systemic and Ileal Markers of Hypoxia

Whole blood analysis revealed that hypoxia increased the number of red blood cells (RBC), haemoglobin (Hb) and haematocrit (Hct) concentrations in WT (*p* = 0.081, *p* < 0.0001 and *p* = 0.0002, respectively) and TNF^∆ARE/+^ mice (*p* = 0.001, *p* < 0.0001 and *p* < 0.0001, respectively) (Figure 2a), indicating a systemic hypoxic state independent of the genotype.

To evaluate whether housing in hypoxic conditions also elicits hypoxia-induced gene expression in the gut, we analysed vascular endothelial growth factor (Vegf) expression. Ileal expression of this hypoxic marker was significantly increased in WT mice housed in hypoxia versus WT mice housed in normoxia (*p* = 0.04; Figure 2b). The mRNA expression of Vegf in TNF^∆ARE/+^ mice housed in hypoxia was increased, but not statistically different from TNF^∆ARE/+^ mice housed in normoxia (*p* = 0.45; Figure 2b).

To confirm and quantify the extent of ileal hypoxia, a pimonidazole staining was performed. Percentual pimonidazole intensity increased significantly in WT mice housed in hypoxia (*p* = 0.017) (Figure 2c–g) and extended beyond the epithelial layer towards the mucosa. In line with colonic inflammation [2,11,19], ileitis increased pimonidazole staining (*p* = 0.039) in normoxic TNF^∆ARE/+^ mice compared with normoxic WT mice. Hypoxic housing conditions tended to further increase the pimonidazole intensity (*p* = 0.08; Figure 2c) and extent (Figure 2d–g).

### 3.3. Environmental Hypoxia Induces Systemic Inflammation in WT Mice

Given the clear presence of both systemic and ileal hypoxia in mice exposed to environmental hypoxia, we questioned whether this was associated with systemic inflammation. Therefore, we first compared the circulatory immune cell abundance in all mice. The number of total white blood cells did not differ between genotypes, nor between housing environments (Appendix A). However, hypoxia significantly increased the number of circulatory monocytes in both TNF^∆ARE/+^ and WT mice (*p* = 0.004 and *p* = 0.038, respectively) (Figure 3a,b). Next to monocytes, TNF^∆ARE/+^ mice housed in hypoxia also exhibited significantly higher numbers of circulatory neutrophils compared with TNF^∆ARE/+^ mice housed in normoxia (*p* = 0.023) (Figure 3b). No significant differences could be identified in the number of lymphocytes, eosinophils and basophils in both TNF^∆ARE/+^ mice and WT littermates independent of their housing environment (Appendix A). Next, we assessed whether the altered immune cell composition is associated with increased proinflammatory cytokine levels. Environmental hypoxia exposure tended to increase TNF (*p* = 0.067), but significantly increased CCL2 (*p* = 0.0158) and IL1β (*p* = 0.0005) levels in the serum of WT mice (Figure 3c). As expected, normoxic TNF^∆ARE/+^ mice exhibited an increase in circulating TNF, CCL2 and IL1β levels (*p* = 0.0297, *p* = 0.0023 and *p* = 0.1039; respectively) compared with normoxic WT littermates (Appendix A). However, hypoxia did not significantly increase these levels in TNF^∆ARE/+^ mice compared with normoxic TNF^∆ARE/+^ mice ((*p* = 0.1127, *p* = 0.6082 and *p* = 0.1675, respectively; Figure 3d).

### 3.4. Environmental Hypoxia Induces a ProInflammatory Microenvironment in the Small Intestine of WT Mice

Since hypoxia induces systemic inflammation in WT mice, we next assessed whether gut homeostasis is affected as well. Histologically, WT mice housed under hypoxic conditions exhibited a trend toward an increased total inflammation score (*p* = 0.066), mainly due to a significant rise in the immune cell infiltration (*p* = 0.048), compared with their normoxic littermate controls (Figure 4a–e and Appendix A). Given the increased immune cell infiltration in hypoxic WT mice, we further characterized this infiltrate by determining the expression of a common mononuclear phagocyte marker Cd11c (alias Itgax) and a lamina propria dendritic cell (DC)-specific marker Cd103 (alias Itgae) [20]. Cd11c expression in the distal ileum of WT mice housed in hypoxia was significantly higher compared with normoxic mice (*p* = 0.0322), while the expression of Cd103 remained unaltered (*p* = 0.655) (Appendix A). This indicates that the increase in mononuclear phagocytes is most likely the result of an increase in a CD11c^+^ macrophage subset. To determine the number of neutrophils in the ileum, we analysed the mRNA levels of the neutrophil marker Ly6G. No significant differences could be identified in Ly6g expression in the distal ileum of mice housed in hypoxia versus normoxia (*p* = 0.0862; Appendix A).

Next, we analysed proinflammatory cytokine expression. In line with the serum levels, hypoxia increased *Tnf* (*p* = 0.038), *Ccl2* (*p* = 0.021) and *Il1β* (*p* = 0.038) expression in the distal ileum of WT mice (Figure 4f). These findings suggest that environmental hypoxia induces an increase in proinflammatory cytokine and chemokine expression.

### 3.5. Environmental Hypoxia Has No Impact on Chronic Ileitis

Given the proinflammatory ileal microenvironment in WT mice exposed to hypoxia, we next questioned whether environmental hypoxia would aggravate the progression of TNF-driven chronic ileitis. On histology, TNF^ΔARE/+^ mice exhibited, as expected, significant villous blunting, shortening and broadening with a massive immune cell infiltration in both mucosa and submucosa, resulting in a significantly higher total inflammation score compared with their WT littermate controls in normoxia (*p* = 0.0003; Appendix A). However, no significant differences in the total inflammation score or subscores could be identified between the TNF^ΔARE/+^ mice housed in hypoxia versus normoxia (Figure 5a–e and Appendix A). This indicates that environmental hypoxia does not protect nor promote chronic ileitis development.

Given the increase in circulatory monocytes and neutrophils, we next investigated whether this leads to increased infiltration in the ileum of hypoxic TNF^∆ARE/+^ mice. As expected, normoxic TNF^∆ARE/+^ mice exhibited a significant increase in Cd11c and Ly6g expression (*p* = 0.0136 and *p* = 0.020, respectively) compared with normoxic WT littermates. The expression of Cd103 was not significantly altered between genotypes (Appendix A). However, hypoxia did not further increase the gene expression of Cd11c and Cd103 (Appendix A), indicating that both macrophages and DCs are not more abundant in the distal ileum of TNF^∆ARE/+^ mice housed in hypoxia versus normoxia. Despite the elevated circulatory neutrophil numbers in TNF^∆ARE/+^ mice housed in hypoxia, Ly6g expression in the distal ileum of TNF^∆ARE/+^ mice was not different depending on their housing environment. However, a downward trend could be identified in the Ly6g expression of TNF^∆ARE/+^ mice housed in hypoxia versus normoxia, indicating an inverse effect between tissue and circulation (Appendix A).

As expected, TNF^∆ARE/+^ mice exhibited a significantly higher ileal expression of the proinflammatory genes, *Tnf* (*p* = 0.009), *Ccl2* (*p* = 0.006) and *Il1β* (*p* = 0.009) compared with normoxic WT mice (Appendix A). Hypoxia exposure did not further increase this proinflammatory expression profile in TNF^∆ARE/+^ mice (Figure 5f). Together, these data demonstrate that chronic environmental hypoxia exposure does not reduce nor increase chronic ileitis development.

### 3.6. Haematopoietic Deletion of Phd1 Does Not Protect against Chronic Ileitis

It has been previously shown that hypoxia exposure suppresses acute colonic inflammation [2]. This hypoxia-induced protection may rely on its inhibitory action on Phd1 activity, since Phd1-deficient mice are protected from acute colitis. To assess whether hypoxia already impacts PHD1 expression, ileal Phd1 mRNA levels were compared between WT and TNF^∆ARE/+^ mice housed in hypoxia and their respective normoxic controls. However, no significant difference could be identified, indicating that hypoxia does not influence Phd1 expression levels (Appendix A). Despite the lack of effect on Phd1 expression, our group previously showed that the deletion of Phd1 in all immune cells (using Phd1^f/f^Vav:cre^+/−^ mice) is necessary and sufficient to elicit a protective effect on acute colitogenesis [11]. However, its effect on chronic ileitis has never been studied. Although our results show that hypoxia does not impact ileitis nor ileal Phd1 expression, it does evoke a huge systemic effect that can overrule a potential beneficial effect of selective and cell-specific PHD targeting. Given the promising role of Phd1-deletion on colonic inflammation [9,11], we investigated the effect of immune-cell specific Phd1-deletion on chronic ileal inflammation. We therefore backcrossed Phd1^f/f^Vav:cre^+/−^ mice with TNF^ΔARE/+^ mice. Genotyping results and confirmation of Phd1-deletion in the ileum is described in supplementary results and visualized in Appendix A while Appendix A illustrates the primer binding sites. In a first set-up, weight of their heterozygous offspring was monitored until week 17. No differences in body weight evolution could be observed between all groups (Figure 6a). In addition, no histological differences could be identified between Phd1^f/+^Vav:cre^−/−^TNF^ΔARE/+^ mice and Phd1^f/+^Vav:cre^+/−^ TNF^ΔARE/+^ mice (Figure 6b and Appendix A). Since heterozygous Phd1-deletion is probably insufficient to obtain a phenotype, subsequent interbreedings were set-up to obtain homozygous haematopoietic Phd1-deficient TNF^ΔARE/+^ mice (Phd1^f/f^Vav:cre^+/−^ TNF^ΔARE/+^), which were compared with their Phd1-floxed TNF^ΔARE/+^ counterparts (Phd1^f/f^Vav:cre^−/−^ TNF^ΔARE/+^). Again, weight gain was not significantly different between all genotypes (Figure 6c). As expected, TNF^ΔARE/+^ mice exhibited villous blunting, shortening and broadening with a massive immune cell infiltration in both mucosa and submucosa compared with their WT littermate controls. However, Phd1^f/f^Vav:cre^+/−^ TNF^ΔARE/+^ mice had no improved histological inflammation score compared with the TNF^ΔARE/+^ mice without Phd1-deletion (Figure 6d–f, Appendix A).

Since TNF^ΔARE/+^ mice exhibit increased numbers of ileal mononuclear phagocytes, neutrophils and elevated proinflammatory cytokine/chemokine expression, we next assessed whether these parameters were impacted after Phd1-deletion. As expected and in accordance with the results from above, serum CCL2 (*p* = 0.026) and the mRNA expression of Cd11c (*p* = 0.031), Ly6g (*p* = 0.006), *Tnf* (*p* = 0.035), *Ccl2* (*p* = 0.003) and *Il1β* (*p* = 0.031) was significantly higher in the distal ileum of Phd1^f^^/f^Vav:cre^−/−^TNF^ΔARE/+^ mice compared with Phd1 floxed WT controls. The expression of Cd103 was not significantly altered. No significant differences in immune cell nor proinflammatory gene expression profile could be identified between Phd1^f/f^Vav:cre^+/−^ TNF^ΔARE/+^ and Phd1^f/f^Vav:cre^−/−^ TNF^ΔARE/+^ mice (Appendix A and Figure 7, respectively), indicating that hypoxia-induced signalling, as a result of Phd1-deletion, does not have an effect on ileitis initiation and progression.

## 4. Discussion

The present study demonstrates that environmental hypoxia induces systemic and ileal adaptation, based on an increase in Hct, Hb, ileal pimonidazole intensity and Vegf expression. Hypoxia also induced systemic inflammation, evident from an increased number of circulatory monocytes and serum proinflammatory cytokine levels in WT mice, which was further associated with a higher intestinal inflammation score as a result of elevated immune cell numbers, coinciding with increased proinflammatory gene expression. However, this hypoxia-induced proinflammatory microenvironment did not aggravate the progression of small intestinal inflammation in TNF^ΔARE/+^ mice. Similarly, the genetic targeting of Phd1 in haematopoietic cells did not impact ileal inflammation in this TNF-induced ileitis mouse model.

To investigate the impact of long-term environmental hypoxia and haematopoietic Phd1-deletion on experimental Crohn’s like ileitis, we made use of TNF^ΔARE/+^ mice. The histological onset of terminal ileitis in TNF^ΔARE/+^ mice occurs around 4 weeks of age [14,16]. We exposed the mice to environmental hypoxia starting at 5 weeks of age to minimize the risk of early death as a result of weight loss during the acclimatization period. Weight loss has also been described in high altitude mountaineers [21,22,23,24]. Indeed, all mice housed in hypoxia experienced body weight loss during the first 5 days of the experiment, which was due to loss of appetite and in accordance with previous reports [21,22], while no effect on further weight gain could be observed.

Although no weight evolution differences could be identified, mice housed in hypoxia activated adaptation mechanisms to cope with this low oxygen tension by increasing Hb and Hct concentrations. This is in line with data observed in mice [25,26,27], nonhuman apes [28] and high altitude mountaineers [22,28,29] exposed to environmental hypoxia, and indicate a systemic hypoxic state. Beside systemic effects, previous studies already reported that a shorter hypoxia exposure regime of 18 h at 8% oxygen level also induces colonic hypoxia which beneficially impacts distal colitis [2,26,27]. We demonstrated for the first time that long-term housing at the same oxygen level induces ileal hypoxia, based on an increase in pimonidazole intensity and Vegf expression in both WT and TNF^ΔARE/+^ mice. Pimonidazole staining was already higher in normoxic TNF^ΔARE/+^ mice due to inflammation, which is in accordance with previous observations during colonic inflammation after DSS or 2,4,6-trinitrobenzene sulphonic acid (TNBS) administration [15,30]. Environmental hypoxia had a synergistic effect on pimonidazole staining.

Since we could clearly demonstrate systemic and ileal adaptation upon hypoxic exposure, we next evaluated the presence of systemic signs of inflammation. Interestingly, we found that the number of circulatory monocytes was significantly elevated in all mice housed in hypoxia. This effect might rely on the hypoxia-induced reprogramming to glycolysis, which has been shown to prolong circulatory monocyte survival [25,31,32]. Concomitantly, an increase in serum TNF, CCL2 and IL1β levels could be detected. Although the exact cellular source of these cytokines and chemokine in this context remains speculative, several studies established a direct link between hypoxia and upregulation of these proinflammatory cytokines [33,34,35], which is in line with our findings. Additionally, an increase in circulatory neutrophils could be identified in TNF^ΔARE/+^ mice housed in hypoxia. This observation is in line with the publication of Alvarez–Martins and colleagues [25] in which they showed that hypoxia induces vascular remodelling in the bone marrow during a chronic intermittent hypoxia (CIH) model, resulting in not only an up-regulation of circulatory monocytes but also neutrophils [25]. This elevation under hypoxic conditions was also reported in in vitro studies [36,37,38,39]. In particular, hypoxia exposure is able to reduce neutrophil apoptosis via the activation of hypoxia-inducible factor 1α (HIF1-α) and augments the phosphoinositide 3 kinase γ (PI3Kγ)-dependent neutrophils degranulation, leading to the enhanced release of elastase, myeloperoxidase, lactoferrin and matrix metalloproteinase-9 [38,39,40] and could therefore explain the elevated circulatory neutrophil number in hypoxic TNF^ΔARE/+^ mice. Despite the hypoxia-induced elevated monocyte and neutrophil numbers, serum levels of proinflammatory cytokine were not significantly increased in hypoxic TNF^∆ARE/+^ mice.

In addition to the systemic signs of inflammation, hypoxia elicited ileal inflammation as well, evident from an increase in total inflammation score due to elevated immune cell infiltration. When further characterizing this infiltrate, we showed that hypoxia induced an up-regulation of mononuclear phagocytes and proinflammatory genes in the distal ileum, indicating for the first time that hypoxia exposure in itself has not only a systemic and local effect in the distal colon [2,26,27], but also creates a proinflammatory microenvironment in the distal ileum.

While these data suggest that hypoxia would sensitize toward an additional proinflammatory trigger, we further demonstrated that long-term environmental hypoxia exposure did not promote ileitis progression in TNF^ΔARE/+^ mice, in contrast to previous studies. In particular, Vavricka and colleagues [3] were the first to report that aircraft travel and journeys at regions lying at an altitude >2000 m above sea level increases the risk of flares in IBD patients in remission, suggesting that environmental hypoxia has a harmful effect on the IBD course. However, these data were solely based on a questionnaire while no objective biochemical parameters were assessed. In a prospective study of Cosin–Roger and colleagues [2], short-term environmental hypoxia exposure had a protective effect on acute colitis through NLRP3/mTOR downregulation and autophagy activation while lower levels of proinflammatory cytokines and chemokines were detected in colonic biopsies from CD patients after 3 h in a hypobaric pressure chamber [2]. However, during chronic colonic infection induced by *Citrobacter rodentium*, hypoxia exposure exacerbated clinical symptoms and pathological damage by down-regulating Th1 and Th17 responses [26,27], emphasizing the context-dependent effects of environmental hypoxia. Given the lack of effect of hypoxia on ileitis, this raises the question whether hypoxia-induced signalling pathways play a major role in ileal inflammatory disease. However, environmental hypoxia causes a huge systemic effect with inhibition of all PHD isoforms. Cell-specific and PHD isoform specific targeting may present a better therapeutic approach.

In this context, our group previously showed that haematopoietic deletion of Phd1 is both necessary and sufficient to protect against experimental colitis, which relies, at least in part, on the promotion of macrophages to the anti-inflammatory M2 phenotype [10,11]. Considering an imbalance towards proinflammatory M1 macrophages hallmarks CD, we investigated whether heterozygous and homozygous haematopoietic deletion of Phd1 in TNF^ΔARE/+^ mice would have a protective effect on experimental Crohn’s like ileitis development. First, heterozygous haematopoietic deletion of Phd1 in TNF^ΔARE/+^ mice did not impact the body weight evolution nor improve histological inflammation compared with TNF^ΔARE/+^ mice without heterozygous haematopoietic Phd1-deletion. This is perhaps not too surprising since heterozygosity of Phd1 has not shown a phenotype in other models of disease. However, homozygous haematopoietic deletion of Phd1 in TNF^ΔARE/+^ mice did not cause a protective effect on the initiation and course of ileitis either. Combined with the lack of effect after hypoxic exposure, these data suggest that hypoxia-induced signalling pathways do not have an effect on the initiation and course of experimental Crohn’s like ileitis.

This is in contrast with a previous publication from our group, where we showed that dimethyloxalylglycine (DMOG)-treated TNF^ΔARE/+^ mice experienced clinical benefit with clear attenuation of chronic intestinal inflammation, associated with HIF1α activation [15]. Although DMOG is considered as a hypoxia-mimetic, it structurally mimics 2-OG and thereby targets not only the catalytic domain of all PHDs and FIH, but also other 2-OG dependent oxygenases by blocking the entry of this cosubstrate [7]. This indicates that DMOG’s activity is not limited to activating hypoxia-induced signalling pathways via PHD/FIH inhibition. Indeed, in inflammation-induced fibrosis development for example, DMOG reduces fibrogenesis via suppression of ERK-mediated TGF-β1 signalling which is not phenocopied in Phd2*^+/−^* mice and independent of HIF-1 activation [41]. These DMOG-induced alternative signalling pathways may be inactivated during long-term environmental hypoxia exposure and haematopoietic Phd1 knock-out, resulting in a lack of protection on ileal inflammation in TNF^ΔARE/+^ mice. In addition, Keely and colleagues [42] demonstrated that the pan-hydroxylase inhibitor AKB-4924 ameliorates colitis with strong evidence for epithelial HIF-1 activation as the driving force. They reported a similar improvement on ileitis in TNF^ΔARE/+^ mice, but the protective mechanism here remains speculative. Although environmental hypoxia also activates HIF-1α, its action is not limited to HIF-1-specific signalling. Hypoxia activates, in addition to HIF-1α-dependent signalling, multiple other signalling pathways as well via the activation of HIF-2 and NFκB in not only epithelial cells, but also in a variety of other cell types [7]. This could explain our observed lack of effect on small intestinal inflammation after hypoxic exposure. Hereby assuming that hydroxylase-targeting in epithelial cells is needed for the protective effect on experimental ileitis, it is not surprising that immune cell-specific Phd1-deletion is insufficient to induce a beneficial effect. In addition, since Phd2 is the main oxygen sensor driving HIF-1 activation [43], the therapeutic effect of AKB-4924 may therefore rely on its Phd2-inhibitory effect which remains to be investigated. To date, several PHD inhibitors have been developed and are currently being tested for various diseases [7]. Vadadustat, Roxadustat and Daprodustat successfully finished phase III trials for the treatment of chronic kidney disease (CKD)-related anemia [44,45,46] and are currently awaiting FDA and EMA approval. Molidustat, another PHD inhibitor, is also being tested in both early and advanced clinical stages for the same indication. The results of three phase III studies are expected soon (NCT033511166, NCT03418168, NCT03543657). So far, only one PHD inhibitor GB004 (formerly AKB-4924, Aerpio therapeutics, Ohio) is currently being tested for the use in IBD. A first phase 1b study in UC patients has been successfully completed (NCT02914262) by Gossamer bio. Results have been presented at the UEGW (United European Gastroenterology Week) congress of 2020, showing tolerance and a gut-targeted PK profile. mRNA profiling on colonic biopsies further revealed target engagement and enhanced epithelial barrier function, resulting in signs of clinical activity compared with placebo [47]. A phase II study in UC patients is currently ongoing (NCT04556383). 

## 5. Conclusions

In summary, we demonstrated for the first time that long-term environmental hypoxia exposure and haematopoietic deletion of Phd1 had no effect on ileal inflammation in a Crohn’s like ileitis mouse model. Hence, hypoxia-induced signalling pathways are most likely not the main drivers of ileitis pathophysiology. In addition, this study suggests that targeting hypoxia-induced signalling, via currently available PHD inhibitors for example, will probably have no beneficial effect in IBD patients with ileal disease. Our findings, together with the data from previous studies, further imply that only patients with colonic disease should be considered in clinical trials with pan-hydroxylase inhibitors. Efficacy studies regarding the use of PHD inhibitors should than also take disease location into consideration as a confounder.

## Figures and Tables

**Figure 1 biology-10-00887-f001:**
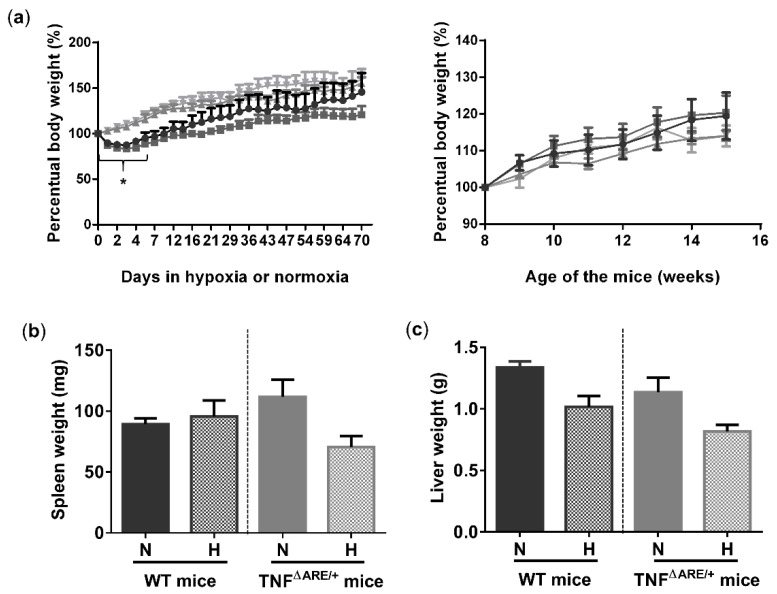
Environmental hypoxia does not influence the general well-being of WT mice and the clinical course of gut inflammation in TNF^∆ARE/+^ mice. (**a**, **left**) Percentual body weight evolution of TNF^∆ARE/+^ mice (
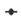
) and their WT littermates (
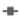
) housed for 10 weeks in hypoxia and TNF^∆ARE/+^ (
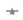
) and WT mice (
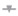
) housed for 10 weeks in normoxia, all from 5 until 15 weeks of age. (**a**, **right**) Percentual body weight evolution of 8 till 15-week-old TNF^∆ARE/+^ mice (
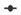
) and their WT littermates (
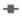
) housed for 10 weeks in hypoxia and TNF^∆ARE/+^ (
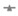
) and WT mice (
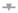
) housed for 10 weeks in normoxia, all from 5 until 15 weeks of age. (**b**) Spleen and (**c**) liver weight of all mice measured at week 15. Data are represented as the mean ± standard error of the mean (SEM). N: Normoxia; H: Hypoxia. * *p* < 0.05.

**Figure 2 biology-10-00887-f002:**
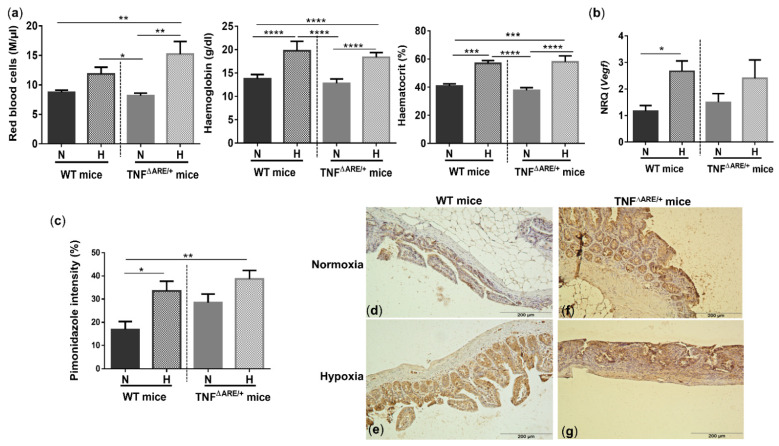
Systemic and ileal hypoxic markers are increased in mice housed in environmental hypoxia. (**a**) Red blood cells, haemoglobin and haematocrit concentrations, (**b**) ileal vascular endothelial growth factor (Vegf) mRNA expression and (**c**) ileal pimonidazole intensity in WT and TNF^∆ARE/+^ mice housed in hypoxia and normoxia. Representative images of the pimonidazole staining in a (**d**) WT mouse housed in normoxia; (**e**) WT mouse housed in hypoxia; (**f**) TNF^∆ARE/+^ mouse housed in normoxia and (**g**) TNF^∆ARE/+^ mouse housed in hypoxia. * *p* < 0.05, ** *p* < 0.01, *** *p* < 0.001, **** *p* < 0.0001. Data are represented as the mean ± SEM. N: Normoxia; H: Hypoxia.

**Figure 3 biology-10-00887-f003:**
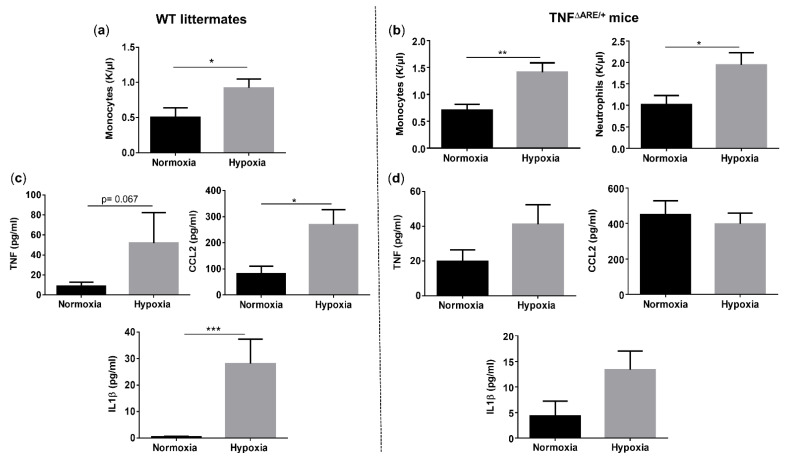
Long-term environmental hypoxia exposure in WT mice induces systemic inflammation. (**a**) Circulatory monocytes were increased in WT mice housed in hypoxia. (**b**) In TNF^∆ARE/+^ mice housed in hypoxia, circulatory monocytes and neutrophils are increased in comparison to TNF^∆ARE/+^ mice housed in normoxia. Serum levels of TNF, CCL2 and IL1β in (**c**) WT littermates housed in normoxia and hypoxia and (**d**) TNF^∆ARE/+^ mice housed in normoxia and hypoxia. * *p* < 0.05, ** *p* < 0.01, *** *p* < 0.001. Data are represented as the mean ± SEM.

**Figure 4 biology-10-00887-f004:**
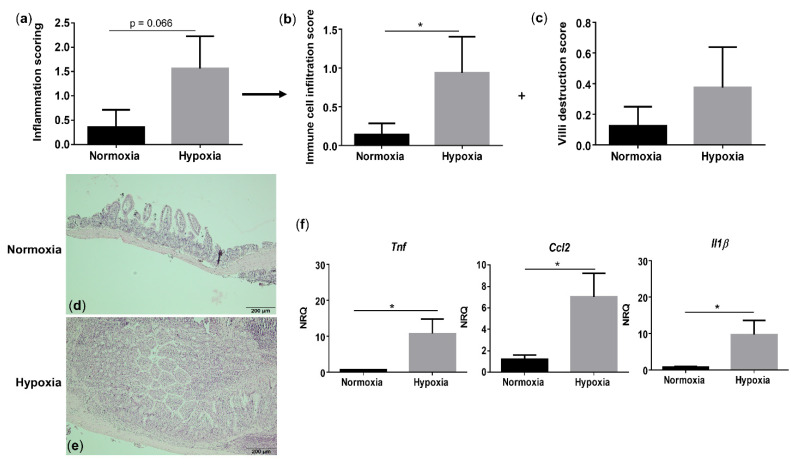
Long-term environmental hypoxia exposure in WT mice induces a proinflammatory ileal microenvironment. (**a**) Total histological inflammation score in WT mice housed in normoxia and hypoxia, comprising the subscores, (**b**) immune cell infiltration and (**c**) villi destruction. Representative images from haematoxylin-eosin (H&E) sections of the distal ileum from (**d**) a WT mouse housed in normoxia and (**e**) a WT mouse housed in hypoxia. (**f**) mRNA expression levels of *Tnf*, *Ccl2* and *Il1β* in WT littermates housed in normoxia and hypoxia. * *p* < 0.05. Data are represented as the mean ± SEM. NRQ: Normalized relative quantities.

**Figure 5 biology-10-00887-f005:**
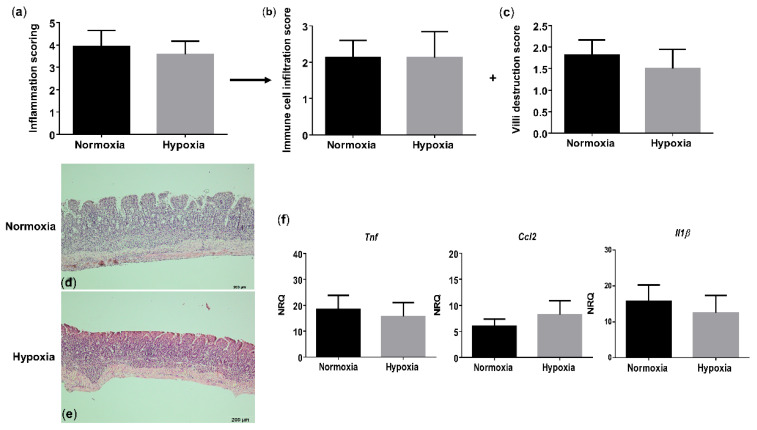
Long-term environmental hypoxia exposure to TNF^ΔARE/+^ mice does not protect against ileitis. (**a**) Total histological inflammation score in TNF^ΔARE/+^ mice housed in normoxia and hypoxia, comprising the subscores: (**b**) immune cell infiltration and (**c**) villi destruction. Representative images from H&E sections of the distal ileum from (**d**) a TNF^ΔARE/+^ mouse housed in normoxia; (**e**) a TNF^ΔARE/+^ mouse housed in hypoxia. (**f**) mRNA expression levels of *Tnf, Ccl2* and *Il1β* in WT littermates housed in normoxia and hypoxia. Data are represented as the mean ± SEM. NRQ: Normalized relative quantities.

**Figure 6 biology-10-00887-f006:**
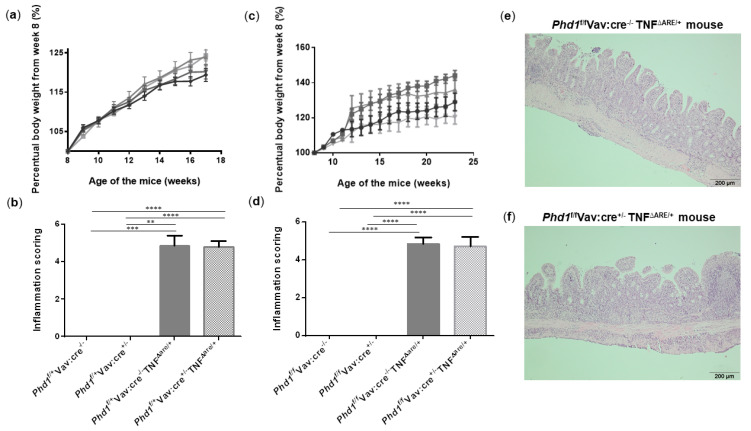
Haematopoietic deletion of Phd1 does not protect against ileitis in TNF^ΔARE/+^ mice. (**a**) Percentual body weight evolution from week 8 till 17 of Phd1^f/+^Vav:cre^+/−^ mice (
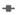
), Phd1^f/+^ Vav:cre^+/−^ TNF^∆ARE/+^ mice (
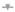
), Phd1^f/+^Vav:cre^−/−^ mice (
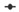
) and Phd1^f/+^Vav:cre^−/−^ TNF^∆ARE/+^ mice (
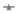
). (**b**) Total histological inflammation scores of the distal ileum from heterozygous haematopoietic Phd1 floxed mice. (**c**) Percentual body weight evolution from week 8 till 23 of Phd1^f/f^Vav:cre^+/−^ mice (
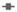
), Phd1^f/f^Vav:cre^+/−^ TNF^∆ARE/+^ mice (
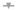
), Phd1^f/f^ Vav:cre^−/−^ mice (
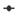
) and Phd1^f/f^Vav:cre^−/−^ TNF^∆ARE/+^ mice (
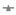
). (**d**) Total histological inflammation scores of the distal ileum from homozygous haematopoietic Phd1 floxed mice. Representative H&E-stained ileal sections from (**e**) a Phd1^f/f^Vav:cre^−/−^ TNF^∆ARE/+^ and (**f**) a Phd1^f/f^Vav:cre^+/−^ TNF^∆ARE/+^ mouse. ** *p* < 0.01, *** *p* < 0.001, **** *p* < 0.0001. Data are represented as the mean ± SEM.

**Figure 7 biology-10-00887-f007:**
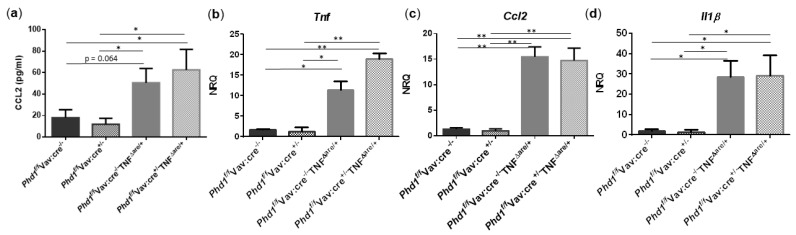
Haematopoietic deletion of Phd1 does not influence the proinflammatory gene expression profile in TNF^ΔARE/+^ mice. (**a**) Serum levels of CCL2 and mRNA expression levels of (**b**) *Tnf,* (**c**) *Ccl2* and (**d**) *Il1β* in the distal ileum of Phd1^f/f^Vav:cre^−/−^, Phd1^f/f^Vav:cre^+/−^, Phd1^f/f^Vav:cre^+/−^TNF^ΔARE/+^ and Phd1^f/f^Vav:cre^+/−^TNF^ΔARE/+^ mice. * *p* < 0.05; ** *p* < 0.01. Data are represented as the mean ± SEM. NRQ: Normalized relative quantities.

## Data Availability

The data presented in this study are readily available upon reasonable request to the corresponding author.

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
