# Peer review of "Long-Term Environmental Hypoxia Exposure and Haematopoietic Prolyl Hydroxylase-1 Deletion Do Not Impact Experimental Crohn’s Like Ileitis"

_biology, 2021, doi:10.3390/biology10090887_

Round 1
Reviewer 1 Report
Authors investigated the effect of long-term exposure to hypoxia on experimental ileitis using Phd1-deletion mouse models. The overall study highlighted that no significant association was noted in hypoxia or haematopoietic Phd1-deletion.
Here are my few comments.
- The graphical abstract needs clear representation of experimental genotypes.
- How long were the mice exposed to hypoxia or normoxia as the abstract states 10 weeks and figure 1 shows 14 weeks, figure 6 shows 18 or 24 weeks. Please clarify.
- As stated in the materials and methods 2.1 sections, did the authors confirmed the Phd1 deletion specifically in all haematopoietic cells of distal ileum. Include the PCR results of phd1 deletion genotypes of the distal ileum.
- With regards to the sample collection of the ileum, did authors collect the whole ileum or distal part of the ileum for the histological studies or the RNA extraction. How was the sample collected for histological studies, whether the tissue was horizontally opened and then rolled to cover the entire region?
- In my opinion, the authors specifically looking for the Phd1 deletion specifically in all haematopoietic cells in distal ileum, why the whole was collected for the RNA extractions?
- Do you have any specific criteria for including the males and females in the study? why the results are not presented between the males and females.
- Apart from H&E and Pimonidazole staining, did the authors confirmed the expression of the Phd1 in all the collected samples?
- It would be more interesting to present the data comparing the heterozygous versus homozygous groups for all the figures. As all the results presented as WT mice and TNF mice.
Reviewer 2 Report
In the manuscript ID biology-1345824 titled “Long-term environmental hypoxia exposure and haematopoietic prolyl hydroxylase-1 deletion do not impact experimental Crohn’s like ileitis” by Cara de Galan and colleagues, the authors have reported the effects of normoxia and hypoxia in five-week-old, TNFΔARE/+ mice and wildtype (WT) littermates were for ten weeks. Although environmental hypoxia increased both systemic as ileal markers of hypoxia, the bodyweight evolution in both WT and TNFΔARE/+ mice was not affected. Interestingly, hypoxia did increase circulatory monocytes, ileal mononuclear phagocytes and pro-inflammatory cytokine expression in WT mice. However, no histological or inflammatory gene expression differences in the ileum could be identified between TNFΔARE/+ mice housed in hypoxia versus normoxia nor between TNFΔARE/+ and WT mice with additional loss of immune cell-specific Phd1 expression. The authors have stated that is the first study showing that long-term environmental hypoxia or haematopoietic Phd1-deletion does not impact experimental ileitis and therefore strongly questions whether targeting hypoxia-induced signalling via currently available PHD inhibitors would exert an immune suppressive effect in IBD patients with ileal inflammation. I have few concerns regarding the present manuscript.
-The authors need to follow the author guidelines regarding references, [].
-I read with interest the manuscript, the authors have done great work, using different techniques in a well-described animal model, however, they need to report some validation techniques using samples to extract proteins to confirm those results, such as, western blot or measuring with an ELISA kit.
-The other sections are well-described and structured to facilitate the reading.
Round 2
Reviewer 2 Report
Thank you to the authors for taking into account my previous comments. All comments were answered correctly in the present manuscript and the manuscript, in general, was improved.